# Epidemiological Study of Physical Activity, Negative Moods, and Their Correlations among College Students

**DOI:** 10.3390/ijerph191811748

**Published:** 2022-09-17

**Authors:** Bo Li, Wen-xia Tong, Meng Zhang, Guang-xu Wang, Yang-sheng Zhang, Shu-qiao Meng, Ya-xing Li, Zhong-lei Cui, Jun-yong Zhang, Yu-peng Ye, Shan-shan Han

**Affiliations:** 1Physical Education College, Shangqiu University, Shangqiu 476000, China; 2Physical Education College, Yangzhou University, Yangzhou 225127, China; 3No. 2 Experimental Primary School in Haidian, Beijing 100085, China; 4College of Physical Education, Henan Normal University, Xinxiang 453007, China; 5School of Physical Education, Nanjing Xiaozhuang University, Nanjing 210000, China; 6Physical Education College, Shangqiu Normal University, Shangqiu 476000, China; 7School of Physical Education, Henan University of Economics and Law, Zhengzhou 450046, China; 8School of Physical Education, Jing-Gang-Shan University, Ji’an 343009, China; 9Institute of Sports Science, Nantong University, Nantong 226019, China

**Keywords:** college students, physical activity, negative moods, mental health, health promotion, epidemiological investigation, static behavior

## Abstract

Objective: Negative moods in college students have caused frequent extreme behaviors. This study analyzed the current status and correlation between physical activity and negative moods in college students. Methods: A cross-sectional study design was used in the present research. Data on college students’ physical activity and negative moods were collected using the Sports Questionnaire Star software. The questionnaires were administered to college students in five colleges and universities in Henan and Jiangsu Provinces, China, and a total of 3711 correctly completed questionnaires were returned. Data on sociological and demographic information, the International Physical Activity Questionnaire Short Form (IPAQ-SF), and the Depression Anxiety and Stress Scale (DASS) were collected. The research was conducted in December 2021. Results: The low-intensity physical activity rate among college students was 55.56%, while depression, anxiety, and stress symptoms were detected in 35.14%, 65.29%, and 10.99%of the college students, respectively. Depression (K = 35.58, *p* < 0.001) and anxiety (K = 15.56, *p* < 0.001) rates were significantly different among the different physical activity intensity groups. The proportion of students who perform high-intensity physical activities was lower than those who perform low- and moderate-intensity physical activities. Conclusion: Low physical activity and high anxiety are evident among college students, and prolonged moderate-to-low-intensity physical activity (including static behavior) induces depression and anxiety. In the future, further studies can be conducted on improving the physical activity intensity of college students, improving the mental health monitoring and intervention systems of college students, and exploring the dose–effect relationship between physical activity and negative moods.

## 1. Introduction

College students are in the transitional stage between adolescence and adulthood, and frequent psychological problems in this group of individuals have been evident in recent years due to external factors, such as interpersonal relationships, study pressure, campus life, and bad cognition. Extreme events, such as suicide and self-harm, are frequent among students [1,2]. Most emotional behavior problems caused by negative moods (NMs) are associated with negative outcomes [3,4]. Negative moods are an individual’s perception of various painful and unpleasant experiences, and negative emotions often lead to engagement in serious self-harmful behaviors, such as suicide, and trigger physiological reactions, such as crying [5,6]. Depression, anxiety, and stress are the three most common negative emotions among college students [7]. Previous research has shown that the interconnection among the three emotions is a common cause of extreme behaviors, such as suicide, among college students [8,9]. Therefore, scholars devoted to research on the mental health of college students have focused on factors that can induce NMs.

Physical activity (PA) refers to activities that increase the body’s energy expenditure due to skeletal muscle contraction [10,11]. Insufficient PA is one of the four major risk factors for noncommunicable diseases and related deaths [10]. Research on exercise and mental health, especially the promotion of physical and mental health from the perspective of PA behavior, has increased rapidly in recent years [12,13]. Studies have shown that an average of 10 regular minutes of PA promotes good physical and mental health outcomes. Moderate-to-vigorous physical activity (MVPA) for any length of time has physical and mental health benefits [14,15,16]. Conversely, prolonged sedentary behavior (SB) and sustained light physical activity (LPA) are some of the predisposing factors for NMs [17,18,19].

The recent decline in PA in different groups, especially the reduction in time spent outdoors due to the global COVID-19 pandemic, is a key cause of NMs [20,21,22,23]. The paradigms and methods on the relationship between PA intensity and NMs used in the current research differ from those of previous studies. Therefore, uncertain, inconsistent, and contradictory conclusions may be drawn [24,25,26]. This study aimed to determine if PA reduces or increases the risk of developing negative emotions.

Exploring the PA and NMs of college students from an epidemiology perspective can effectively show the group characteristics, contrast characteristics, and the overview of engagement in PA and NM rates among college students. Epidemiology is the science of the distribution of diseases and factors influencing this phenomenon. It aims to explore the cause, clarify the epidemic law, and formulate preventive and control measures of a disease [27]. International scholars have gradually recognized the intersection between physical education and epidemiology as an effective way to promote participation in PA. Public health-oriented PA epidemiological studies allow for an in-depth analysis of the factors influencing participation in PA and their correlation with other health behaviors, highlighting the health benefits of large-scale epidemiological investigations [28].

This was a cross-sectional study in which the association between PA and NMs was analyzed using a large-sample epidemiological empirical survey. The purpose of the study was: (1) to investigate the current status of PA among college students from an epidemiological perspective, promote participation in PA among Chinese college students, and provide data support for subsequent research on PA-related paradigms; (2) to investigate the current status of NMs among college students from an epidemiological perspective, explore the current mental health status of college students in China from a large sample perspective, and provide theoretical support for follow-up mental health monitoring and intervention work for college students; (3) to analyze the NM rates among college students under different PA intensities to explore the correlation between PA and negative emotions and to provide theoretical support for the formulation of subsequent joint intervention programs. The findings of this study are significant for preventing negative emotions, such as anxiety, depression, and stress, among college students and promoting physical and mental health development.

## 2. Materials and Methods

### 2.1. Participants

Freshmen (50.66%), sophomores (34.06%), and juniors (15.28%) in five universities: Shangqiu College and Shangqiu Normal College in Shangqiu City, Henan Province; Shanghai University; Yangzhou University in Yangzhou City, Jiangsu Province; and Nantong University in Nantong City were sampled for this study. The study was conducted in December 2021 using the “Questionnaire Star” software. According to the 2020 National Economic and Social Development Statistical Bulletin released by the National Bureau of Statistics [29], the selected colleges and universities in Henan Province are in low socioeconomic areas, while the universities in Jiangsu Province and Shanghai are in high socioeconomic areas. Fourth-year students were not included in this study because they were undertaking their practicums and writing their theses.

The equation shows the minimum sample size required for this study [30]. Type I error α was set to 0.05; the allowable error δ was set to 0.01, and the sample rate ρ was set at 0.05. The official websites of the five colleges and universities indicate that the universities had 154,689 undergraduates (data updated in 2020). Therefore, the finite population N assumes that 75% of the total number of students is about 116,017. The minimum sample size required for this study was *n* = 1797.

A total of 4201 questionnaires were distributed, and after excluding those with missing key information (such as age) and poor or wrong responses, 3711 (88.34%) correctly completed questionnaires were included in the subsequent analyses. The sample size met the minimum sample size requirement. The distribution of the demographic characteristics is shown in Table 1.

### 2.2. Measurements

#### 2.2.1. International Physical Activity Questionnaire Short Form (IPAQ-SF)

Physical activity level was measured using the IPAQ-SF, developed by the International Physical Activity Measurement Working Group in 2001. The IPAQ-SF contains seven questions, six on the individual’s PA. The 7th question inquires about the individual’s sedentary time. The daily cumulative time and one-week frequency of participating in low-intensity PA, moderate-intensity PA (MPA), and vigorous physical activity (VPA) within seven days were recorded.

The IPAQ-SF is a highly reliable and validated tool among Chinese college students for assessing PA [31,32]. The energy consumption for PA was calculated using Equation (1) [33].
***Energy Consumption (Met) = Intensity × Frequency (d/w) × Duration (min/d)***(1)

The energy expenditure of the three intensities was summed to obtain the total PA expenditure for the week. Among them, high-intensity activities were assigned a MET value of 8.0; moderate-intensity activities were assigned a MET value of 4.0, and walking was assigned a value of 3.3. The types of PA were classified into the following groups: LPA: (1) No PA was reported, and (2) the reported PA did not meet the high and medium grouping criteria; MPA: (1) The total number of days of all high-intensity PA was ≥3 days, ≥20 min each session, (2) the total number of days for any kind of moderate-intensity and/or walking activities was ≥5 days, each ≥20 min, (3) intense PA ≥ 5 days/week, and the total weekly PA was ≥600 MET min/w; VPA: (1) The total number of days of any type of high-intensity PA ≥ 3 days, and a weekly PA ≥ 1500 MET min/w, (2) PA days for any of the three intensities ≥ 7 days, with a weekly PA ≥ 3000 MET min/w. SB is classified in the LPA level [33].

#### 2.2.2. Depression Anxiety and Stress Scale (DASS)

NM measurements were performed based on the DASS levels. The internal structure of the DASS is based on the three-dimensional model proposed by Clark and Watson, in which depression and anxiety are thought to have both unique and common symptoms [34]. This study used the ***21-Item Depression Anxiety and Stress Scale (DASS-C21)*** developed by Lovibond et al. and revised by Antony et al. [35] with seven questions for each dimension to assess NMs. The scale uses the Likert 4-point scoring standard, with scores ranging from 0 to 3, with higher scores indicating higher levels of negative emotions.

The reliability and validity of the DASS-C21 for measurement among Chinese college students are high. [35]. The DASS-C21 scores for depression among college students are as follows; ≤9 is normal, 10–13 is mild depression, 14–20 is moderate depression, 21–27 is severe depression, and ≥28 is very severe depression. For anxiety, a score ≤ 7 is considered normal, 8–9 is mild anxiety, 10–14 is moderate anxiety, 15–19 is severe anxiety, and ≥20 is very severe anxiety. For stress, a score of ≤14 is considered normal, 15 to 18 is considered mild stress, 19 to 25 is moderate stress, 26 to 33 is considered severe stress, and ≥34 is considered very severe stress [35]. Negative mood symptoms in clinical practice are also assessed using the DASS-C21 scale.

### 2.3. Quality Control

The quality control of the research was ensured by unifying the research plan, implementing the questionnaire survey, special training of the investigators in the early stage of the survey, developing a standardized introduction language, and ensuring proficiency of the questionnaire content and that the questionnaire had been filled correctly. The investigators were college counselors. Measuring PA and negative emotions is commonly used internationally with high reliability and validity among domestic college students. The questionnaires had to be completed within one week, which reduced selection bias arising from an extended study period. Data preprocessing to correct logic errors and omissions was performed to assure the authenticity and validity of the data. The requirements for statistical data processing were strictly adhered to; corresponding statistical methods were adopted for different data types, and parallelism tests were conducted before performing a generalized linear model test analysis.

### 2.4. Statistical Analysis

Data were analyzed using SPSS 25.0 (SPSS Inc., Chicago, IL, USA) and EXCEL software (Office 2021, Microsoft Corporation, Redmond, WA, USA). Since the data were not normally distributed, they were analyzed using the nonparametric Mann–Whitney U (Mann–Whitney U) test. The differences in PA and negative emotions between genders and socioeconomic levels were analyzed using the nonparametric Kruskal–Wallis H test. The differences in negative emotion levels under different PA intensities were also analyzed using the Kruskal–Wallis H test. The relationship between PA and negative emotion was determined by Spearman’s correlation test. The relative risk of different levels of PA on negative emotions was determined using the generalized linear models. α = 0.05 was considered statistically significant.

## 3. Results

The results of this study are presented from three perspectives: the epidemiological characteristics of negative emotions, the detection of negative emotions under different PA intensities, and the association between PA and negative emotions.

### 3.1. Epidemiological Characteristics of Physical Activity and Negative Moods

The Table 2 shows that the majority of college students (55.36%) engaged in LPA. Depression was detected in 35.14% of college students, of which severe depression or worse was observed in 1.8% of those depressed. Anxiety was detected in 65.29% of the college students, with severe anxiety or worse accounting for 7.63% of the cases. The stress rate among college students was 10.99%, with severe stress or worse accounting for 0.43% of the cases. The PA (Z = −23.96, *p* < 0.001) and anxiety (Z = −4.29, *p* < 0.001) were significantly different between male and female students. In addition, the PA varied among students of different years (K = 34.62, *p* < 0.001), depression levels (K = 37.56, *p* < 0.001), and stress levels (K = 30.41, *p* < 0.001). In addition, PA and NM levels varied among college students of different socioeconomic statuses.

### 3.2. The NM Rates under Different PA Intensities

Table 3 shows the NMs among college students in different PA groups. The depression test showed that the depression rates among college students significantly varied under different PA groups (K = 35.58, *p* < 0.001). Particularly, the depression rates were significantly lower in the LPA (62.71%) and MPA (64.38%) groups. The anxiety rates varied significantly among the different PA levels (K = 15.56, *p* < 0.001). The proportion of students within a normal anxiety range in the LPA group was 32.54% and 31.77% for those in the MPA group, all of which were lower than 43.15% for those in the VPA group. There was no significant difference in stress emotion rates among the different PA groups (*p* = 0.36). Thus, the PA intensity is associated with the development of depression and anxiety.

### 3.3. The Association between PA and NM among College Students

The correlation analysis between PA and negative emotions (Table 4) showed that there was a weak negative correlation between both anxiety (*r* = −0.38) and depression (*r* = −0.36) levels and PA levels. There was no correlation between stress and PA level.

Further, after controlling for intervening variables, including gender, year, and socioeconomic level, generalized linear models were constructed using the PA intensity level as the independent variable and depression and anxiety as the dependent variables. All models passed the Mann–Whitney U test (*p* > 0.05). The results show that compared to VPA, LPA (***β*** = 0.35) and MPA (***β*** = 0.27) increased the risk of depression; anxiety was higher in individuals in the LPA (***β*** = 0.46) and MPA (***β*** = 0.43) groups (Table 5). Therefore, VPA, LPA, and MPA are risk factors for developing depression and anxiety disorders, with LPA being the most significant.

## 4. Discussion

This study showed that most college students (55.36%) mostly engage in mainly LPA, consistent with previous studies [36,37]. The significant decline in engaging in MVPA globally is closely related to changes in nutrition, transportation, and lifestyle [10]. This study was conducted during the COVID-19 pandemic period when government policies restricted outdoor movement and participation in certain PA.

More boys engage in PA than girls (Z = −23.96, *p* < 0.001). Similar studies show that girls are more willing to engage in low-intensity sports, mostly leisure-related sports (e.g., yoga). [33]. More lower-year students than upper-year students participated in PA (K = 34.62, *p* < 0.001), consistent with previous studies [38,39]. College students are in a period of transition from adolescence to adulthood. Some studies have shown that boys and girls before age 12 participate in MVPA, but the proportion decreases as they get older, which may be related to the rapid development of human physiology and psychology during adolescence. [40]. From a socioeconomic perspective, relatively economically stable college students engage in more PA than their opposite colleagues (Z = −11.23, *p* < 0.001). Previous studies have shown that high socioeconomic status groups tend to live and maintain a healthier lifestyle (e.g., physical activity and community involvement), and lifestyles is closely related to engagement in PA [41].

Depression was detected in 35.14% of the college students, with severe or worse depression accounting for 1.8% of this. Anxiety was detected in 65.29% of the college students, in which severe or worse anxiety accounted for 7.63% of this; stress was detected in 10.99% of the college students, with severe stress or worse accounting for 0.43% of this. There were differences in depression (Z = −1.33; *p* = 0.18) and anxiety (Z = −4.29; *p* < 0.001) between boys and girls, and the rates were lower in boys than in girls. A similar trend was observed for depression (K = 37.56; *p* < 0.001) and stress (K = 30.41; *p* < 0.001). Previous studies have shown that the depression rate among college students in China is between 8% and 74% [42,43,44], while anxiety is between 5% and 40% [43,44].

There are significant differences in the NM rate among college students in different studies, which may be related to the cultural backgrounds, the research paradigm adopted by the researchers, and the measurement tools [43]. The depression rate among college students reported in this study was consistent with the previous research. Of note is that there is need to pay attention to college students with severe or worse depression (1.8%). The high anxiety rate among college students may be because it was conducted during the end of the semester examination period when students were under pressure of academic assessment. Previous studies have shown that academic stress causes anxiety [45,46]. This study was also conducted during the COVID-19 pandemic when NMs, such as anxiety and depression, were common [20,47].

This study also showed that depression (Z = −7.63; *p* < 0.001), anxiety (Z = −5.67; *p* < 0.001), and stress (Z = −2.95; *p* < 0.001) significantly varied among students from different socioeconomic backgrounds. The depression and stress rates also varied between genders. In addition, negative emotions were higher in college students from middle and low economic backgrounds than in those in the middle to high economic backgrounds, inconsistent with previous studies. An earlier meta-analysis showed that low socioeconomic status (SES) was often associated with higher rates of psychiatric morbidity, disability, and poor access to health care [48]. However, Zimmerman’s research shows that for the young, instrumental variable estimates suggest that financial stress may not cause depression [49]. This study investigated students below the third year who face little employment pressure. In addition, the economic pressure faced by college students may not be related to the region’s socioeconomic level, which may be due to the reasons mentioned above.

This study revealed an inverse correlation between the PA level and the risk of developing depression and anxiety. Despite our findings, the specific dose–response relationship between PA intensity and NMs needs further investigation. Huang et al. demonstrated a strong correlation between LPA and the occurrence of depression (OR = 1.47), but intervening variables (such as self-compassion) influenced this association [50]. An epidemiological survey of junior high school students also found that the risk of NMs was generally lower among students who performed more than 60 min of moderate-intensity PA, compared to students who did not engage in PA for a 1 week (static behavior). The risk of four types of NMs, including anxiety and depression, is significantly lower among students who perform PA for more than three days in a week [51]. Gerber’s research showed that vigorous to intense exercises better impact mental health than moderate PA [52].

In IPAQ, SB was included in LPA for analysis, which might partly explain the higher LPA in this study. Previous studies show that SB is a key factor that induces negative emotions in college students. For example, Gerber et al. reported that the rate of negative emotions among undergraduate students was significantly low in those engaging in VPA than in those that did not (SB) [52]. Herbert et al. showed a significant improvement in self-reported depression, overall perceived stress, and perceived stress due to uncertainty after short aerobic exercises by the study participants [53]. Thus, exercise promotes better mental health.

Regarding limitations, firstly, this cross-sectional study only analyzed the association between PA and ME. Further studies are needed to reveal the dose–response relationship. Secondly, the PA levels were self-reported through questionnaires, and the inclusion of SB in the IPAQ LPA scale may have overestimated the LPA. In addition, objective tools, such as triaxial accelerometers, were not used, which may have led to methodological bias.

## 5. Conclusions

Very few college students engage in PA, particularly girls and senior students. In addition, the socioeconomic level of the region where the university is located is also one of the factors influencing engagement in PA, which influences the development of NMs. Moderate-to-low-intensity PA (including static behaviors) increases the risk factor for the development of negative emotions, such as depression and anxiety.

## Figures and Tables

**Table 1 ijerph-19-11748-t001:** The demographic characteristics of the study participants.

		*n*	%
Sex			
	Male	1697	45.73
	Female	2014	54.27
Year			
	First	1880	50.66
	Second	1264	34.06
	Third	567	15.28
Socioeconomic level			
	High	1996	53.80
	Low	1715	46.20

**Table 2 ijerph-19-11748-t002:** The PA and NM characteristics of the college students (*n* = 3711).

		Physical Activity				NM—Depression				NM—Anxiety			NM—Stress		
		Rank	*n*	%		Rank	*n*	%		*n*	%		*n*	%	
**Overall**	*n* = 3711														
		LPA	2062	55.56		normal	2407	64.86		1288	34.71		3303	89.01	
		MPA	831	22.39		mild	817	22.02		1024	27.59		295	7.95	
		VPA	818	22.04		moderate	420	11.32		1116	30.07		97	2.61	
						severe	51	1.37		213	5.74		16	0.43	
						very serious	16	0.43		70	1.89				
**Sex**															
	Male	*n* = 1697			Z = −23.96 *p* < 0.001				Z = −1.33 *p* = 0.18			Z =−4.29 *p* < 0.001			Z = −0.79 *p* = 0.43
		LPA	603	35.53		normal	1131	66.65		668	39.36		1505	88.69	
		MPA	469	27.64		mild	329	19.39		436	25.69		122	7.19	
		VPA	625	36.83		moderate	193	11.37		454	26.75		55	3.24	
						severe	30	1.77		94	5.54		15	0.88	
						very serious	14	0.82		45	2.65				
	Female	*n* = 2014													
		LPA	1459	72.44		normal	1276	63.36		620	30.78		1798	89.28	
		MPA	362	17.97		mild	488	24.23		588	29.20		173	8.59	
		VPA	193	9.58		moderate	227	11.27		662	32.87		42	2.09	
						severe	21	1.04		119	5.91		1	0.05	
						very serious	2	0.10		25	1.24				
**Year**															
	First	*n* = 1880			K = 34.62 *p* < 0.001				K = 37.56 *p* < 0.001			K = 5.94 *p* = 0.051			K = 30.41 *p* < 0.001
		LPA	1124	59.79		normal	1289	68.56		654	34.79		1720	91.49	
		MPA	380	20.21		mild	409	21.76		555	29.52		128	6.81	
		VPA	376	20.00		moderate	166	8.83		549	29.20		32	1.70	
						severe	16	0.85		106	5.64				
						very serious				16	0.85				
	Second	*n* = 1264													
		LPA	626	49.53		normal	754	59.65		433	34.26		1079	85.36	
		MPA	299	23.66		mild	282	22.31		312	24.68		126	9.97	
		VPA	339	26.82		moderate	185	14.64		398	31.49		47	3.72	
						severe	31	2.45		74	5.85		12	0.95	
						very serious	12	0.95		47	3.72				
	Third	*n* = 567													
		LPA	312	55.03		normal	364	64.20		201	35.45		504	88.89	
		MPA	152	26.81		mild	126	22.22		157	27.69		41	7.23	
		VPA	103	18.17		moderate	69	12.17		169	29.81		18	3.17	
						severe	4	0.71		33	5.82		4	0.71	
						very serious	4	0.71		7	1.23				
**Socioeconomic level**															
	High	*n* = 1996													
		LPA	913	45.74	Z = −11.23 *p* < 0.001	normal	1209	60.57	Z = −7.63 *p* < 0.001	593	29.71	Z = −5.67 *p* < 0.001	1747	87.53	Z = −2.95 *p* < 0.001
		MPA	589	29.51		mild	496	24.85		546	27.35		193	9.67	
		VPA	494	24.75		moderate	255	12.78		689	34.52		49	2.45	
						severe	29	1.45		137	6.86		7	0.35	
						very serious	7	0.35		31	1.55				
	Low	*n* = 1715													
		LPA	1149	67.00		normal	1198	69.85		695	40.52		1556	90.73	
		MPA	242	14.11		mild	321	18.72		478	27.87		102	5.95	
		VPA	324	18.89		moderate	165	9.62		427	24.90		48	2.80	
						severe	22	1.28		76	4.43		9	0.52	
						very serious	9	0.52		39	2.27				

**Table 3 ijerph-19-11748-t003:** NM rates in different PA groups (*n* = 3711).

		LPA (*n* = 2062)		MPA (*n* = 831)		VPA (*n* = 818)	
		*n*	%	*n*	%	*n*	%
Depression							
	normal	1293	62.71	535	64.38	579	70.78
	mild	480	23.28	189	22.74	148	18.09
	moderate	253	12.27	92	11.07	75	9.17
	severe	31	1.50	10	1.20	10	1.22
	very serious	5	0.24	5	0.60	6	0.73
	K	35.58					
	*p*	<0.001					
Anxiety							
	normal	671	32.54	264	31.77	353	43.15
	mild	565	27.40	239	28.76	220	26.89
	moderate	672	32.59	252	30.32	192	23.47
	severe	118	5.72	59	7.10	36	4.40
	very serious	36	1.75	17	2.05	17	2.08
	K	15.56					
	*p*	<0.001					
Stress							
	normal	1831	88.80	733	88.21	739	90.34
	mild	170	8.24	70	8.42	55	6.72
	moderate	56	2.72	23	2.77	18	2.20
	severe	5	0.24	5	0.60	6	0.73
	K	2.04					
	*p*	0.36					

**Table 4 ijerph-19-11748-t004:** Correlation between PA and NMs of college students (*n* = 3711).

	Anxiety	Stress	Physical Activity
Depression	0.71 **	0.48 **	−0.38 **
Anxiety		0.51 **	−0.36 **
Stress			−0.01

** *p* < 0.01.

**Table 5 ijerph-19-11748-t005:** Generalized linear model parameter estimation (*n* = 3711).

		LPA	MPA
Depression			
	* **β** *	0.35	0.27
	Standard error	0.88	0.10
	Wald *x*^2^	15.88	6.90
	* **p** *	<0.001	0.009
Anxiety			
	* **β** *	0.46	0.43
	Standard error	0.76	0.90
	Wald *x*^2^	31.90	25.42
	* **d** *	<0.001	<0.001

## Data Availability

The raw data supporting the conclusions of this article can be made available by the authors, without undue reservation.

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
