# Peer review of "Epidemiological Study of Physical Activity, Negative Moods, and Their Correlations among College Students"

_ijerph, 2022, doi:10.3390/ijerph191811748_

Round 1

Reviewer 1 Report

The authors have written an interesting paper. It has an enormous sample (at least, I found it to be enormous), so it really adds to the strength of the findings. I furthermore appreciate the authors using non-parametric tests, even though in such large sample assumptions of linearity really can be bend.

However, then the authors mention “The relative risk of different levels of PA on negative emotions was determined using the generalized linear models.” Which is odd, since normality is also an assumption of GLM’s. Why was this then done?

Sometimes the language can be clearer. E.g. “The high anxiety rate among college students may be because it was conducted during the end of the semester examination period when students were under pressure to grasp the taught concept.” What taught concept?

Also, sometimes a lot of repetition: “However, Zimmerman’s research shows that for young, recently em- 278 ployed adults, lack of income and financial stress are associated with depression. However, instrumental variable estimates suggest that financial stress may not cause depression[49].”

The conclusion of the article is quite odd: it mentions limitations, despite that this should be done in the discussion section.

“Thus, exercise promotes better mental health.” Yes, but also the other way around.

In general, I would like to see a discussion of something cross-sectional studies on mental health always ignore: are we not medicalizing normal behavior? The authors themselves acknowledge that this was done at the end of a semester. In all honesty, I myself would probably have scored high on depression, anxiety and stress scales during my finals and at the end of a semester. That does not mean I have depression, anxiety or stress. So, to conclude that interventions are needed based on this study alone, is somewhat bluntly. I think the authors should expand a bit more on this in a limitation section in the discussion section. I would personally include something on the importance of time and context when assessing mental health. For example, someone whose friend just died will score positive on a depression scale, yet, might be “cured” after a month. Feeling bad can be normal.

But overall, a nice study. 

Author Response

Point-by-point Responses to Reviewer 1

Dear reviewer,

Thank you for the time and effort that you have dedicated to providing your insightful and valuable comments on our manuscript. Although I do not know the situation around you, please stay healthy and keep safe.

Sincerely,

Comment 1:

However, then the authors mention “The relative risk of different levels of PA on negative emotions was determined using the generalized linear models.” Which is odd, since normality is also an assumption of GLM’s. Why was this then done?

Response 1:

Thank you for your comment. The reason for choosing the generalized linear model is mainly based on the following points. First, if we want to extend the regression method to a nonparametric method for non-normally distributed data, we need to use a generalized linear model. Linear models are suitable for normally distributed continuous variables. But in many cases, it is not reasonable to assume that the dependent variable is normally distributed (even continuous). Generalized linear models extend the framework of linear models to include the analysis of non-normal dependent variables. Secondly, the generalization of the generalized linear model is reflected in its wider range of assumptions and applicable scenarios, and can include models such as linear regression and logistic regression. Therefore, we believe that applying generalized linear models has methodological advantages. Finally, the amount of data in this study is huge, and the statistical methods of general nonlinear models may increase the probability of alpha errors.

Comment 2:

Sometimes the language can be clearer. E.g. “The high anxiety rate among college students may be because it was conducted during the end of the semester examination period when students were under pressure to grasp the taught concept.” What taught concept?

Response 2:

Thank you for your comment. Maybe there is something wrong with my statement. What I want to express is that the assessment at the end of the semester is the main academic pressure for college students, and this pressure may be one of the reasons for anxiety. We have made changes in the article. At the same time, we read the article again and revised some vague statements.

Comment 3:

Also, sometimes a lot of repetition: “However, Zimmerman’s research shows that for young, recently em- 278 ployed adults, lack of income and financial stress are associated with depression. However, instrumental variable estimates suggest that financial stress may not cause depression[49].”

Response 3:

Thank you for your comment. We have made changes. At the same time, we read the article again and revised some vague statements.

Comment 4:

The conclusion of the article is quite odd: it mentions limitations, despite that this should be done in the discussion section.

Response 4:

Thank you for your comment. This is an important reminder. We've made the changes as you suggested. It may be that I am used to writing articles in Chinese.

Comment 5:

In general, I would like to see a discussion of something cross-sectional studies on mental health always ignore: are we not medicalizing normal behavior? The authors themselves acknowledge that this was done at the end of a semester. In all honesty, I myself would probably have scored high on depression, anxiety and stress scales during my finals and at the end of a semester. That does not mean I have depression, anxiety or stress. So, to conclude that interventions are needed based on this study alone, is somewhat bluntly. I think the authors should expand a bit more on this in a limitation section in the discussion section. I would personally include something on the importance of time and context when assessing mental health. For example, someone whose friend just died will score positive on a depression scale, yet, might be “cured” after a month. Feeling bad can be normal.

Response 5:

wow! that's great! Very nice review! I totally agree with your comment! As you say, medicalizing normal behavior is really overkill. In fact, what I want to express is that in the midst of the COVID-19 pandemic, such negative emotions may be more serious. But in any case, moderate and severe negative emotions must attract our attention. So the significance of this article is highlighted. What is the physical condition of college students with moderate and severe negative emotions? This is also a point we focus on in the discussion. Hope my reply will clear up your confusion.

Thank you so much for letting me learn a lot of new knowledge!

Reviewer 2 Report

The authors undertook their research on an important topic, which is the relationship between physical activity and mood in adolescents in the population of almost 4,000 students of 5 Chinese universities. This study is of great importance, especially in the context of the COVID-19 pandemic, which caused a significant reduction in physical activity in people of all ages.

Please find my comments below.

Introduction:

Lines 69-73: I wouldn’t start this paragraph with definition of epidemiology. Please reverse the order of sentences – p.ex. write about relation of PA and NM from epidemiological point of view which could be defined as..

Material and methods:

Line 98: please add grades (as they were mentioned in Table 1) in parenthesis: “Freshmen, sophomores, and juniors”

Line 115: equation is not necessary here – reference [30] in line 108 is enough, please delete

Lines 124-146: please add a reference to the specific ranges of MET values considered as LPA, MPA, etc. This information can be added as a separate sentence.

Conclusions:

Line 313-314: The authors rightly note that adding objective methods to assess subjects’ physical activity (triaxial accelerometers) would provide more reliable results. Nevertheless, for such a large population it would be very difficult to do. I suggest adding this information as "Study limitations" section in "Discussions", not in “Conclusion” section.

Author Response

Point-by-point Responses to Reviewer 2

Dear reviewer,

Thank you for the time and effort that you have dedicated to providing your insightful and valuable comments on our manuscript. Although I do not know the situation around you, please stay healthy and keep safe.

Sincerely,

Comment 1:

Lines 69-73: I wouldn’t start this paragraph with definition of epidemiology. Please reverse the order of sentences – p.ex. write about relation of PA and NM from epidemiological point of view which could be defined as..

Response 1:

Thank you for your comment. This is a very good suggestion. We all accept your suggestions.

Comment 2:

Line 98: please add grades (as they were mentioned in Table 1) in parenthesis: “Freshmen, sophomores, and juniors”

Response 2:

Thank you for your comment. This is a very good suggestion. We all accept your suggestions.

Comment 3:

Line 115: equation is not necessary here – reference [30] in line 108 is enough, please delete

Response 3:

This is a very good suggestion. We have removed the equation.

Comment 4:

Lines 124-146: please add a reference to the specific ranges of MET values considered as LPA, MPA, etc. This information can be added as a separate sentence.

Response 4:

This is a very good suggestion. We've made changes to your suggestion. The specific division of intensity has made detailed standards.

Comment 5:

Line 313-314: The authors rightly note that adding objective methods to assess subjects’ physical activity (triaxial accelerometers) would provide more reliable results. Nevertheless, for such a large population it would be very difficult to do. I suggest adding this information as "Study limitations" section in "Discussions", not in “Conclusion” section.

Response 5:

Thank you for your comment. This is very good suggestion. We've made the changes as you suggested.